# Evaluation of Volatile Organic Compounds Obtained from Breath and Feces to Detect *Mycobacterium tuberculosis* Complex in Wild Boar (*Sus scrofa*) in Doñana National Park, Spain

**DOI:** 10.3390/pathogens9050346

**Published:** 2020-05-02

**Authors:** Pauline Nol, Radu Ionescu, Tesfalem Geremariam Welearegay, Jose Angel Barasona, Joaquin Vicente, Kelvin de Jesus Beleño-Sáenz, Irati Barrenetxea, Maria Jose Torres, Florina Ionescu, Jack Rhyan

**Affiliations:** 1Centers for Epidemiology and Animal Health, Veterinary Services, Animal and Plant Health Inspection Service, United States Department of Agriculture, Fort Collins, CO 80526, USA; 2Department of Electronics, Electrical and Automatic Engineering, Rovira i Virgili University, 43007 Tarragona, Spain; radu.ionescu@urv.cat (R.I.); irati.barrenetxea92@gmail.com (I.B.); florina.ionescu@urv.cat (F.I.); 3The Ångström Laboratory, Division of Solid State Physics, Department of Materials Science and Engineering Sciences, Uppsala University, 75121 Uppsala, Sweden; tesfalem.welearegay@angstrom.uu.se; 4VISAVET Health Surveillance Centre, Animal Health Department, Faculty of Veterinary Medicine, Complutense University of Madrid, 28040 Madrid, Spain; joseangel.barasona@gmail.com; 5SaBio Instituto de Investigación en Recursos Cinegéticos IREC, ETSIA Ciudad Real, University Castilla La Mancha & CSIC, 13003 Ciudad Real, Spain; joaquin.vicente@uclm.es; 6Faculty of Engineering, Universidad Autónoma del Caribe, Barranquilla 080020, Colombia; rikelvin1984@gmail.com; 7Department of Chemical Engineering, Complutense University of Madrid, 28040 Madrid, Spain; 8Biomedical Institute of Sevilla (IBiS), University of Seville, University Hospital Virgen del Rocío/CSIC, 41071 Seville, Spain; mjtorres@us.es; 9National Veterinary Services Laboratory, Veterinary Services, Animal and Plant Health Inspection Service, United States Department of Agriculture, Fort Collins, Colorado, Fort Collins, CO 80521 USA; rhyanjack@yahoo.com

**Keywords:** volatile organic compounds, VOC, *Mycobacterium tuberculosis* complex, wild boar, swine, *Sus scrofa*

## Abstract

The presence of *Mycobacterium tuberculosis* complex (MTBC) in wild swine, such as in wild boar (*Sus scrofa*) in Eurasia, is cause for serious concern. Development of accurate, efficient, and noninvasive methods to detect MTBC in wild swine would be highly beneficial to surveillance and disease management efforts in affected populations. Here, we describe the first report of identification of volatile organic compounds (VOC) obtained from the breath and feces of wild boar to distinguish between MTBC-positive and MTBC-negative boar. We analyzed breath and fecal VOC collected from 15 MTBC-positive and 18 MTBC-negative wild boar in Donaña National Park in Southeast Spain. Analyses were divided into three age classes, namely, adults (>2 years), sub-adults (12–24 months), and juveniles (<12 months). We identified significant compounds by applying the two-tailed statistical t-test for two samples assuming unequal variance, with an α value of 0.05. One statistically significant VOC was identified in breath samples from adult wild boar and 14 were identified in breath samples from juvenile wild boar. One statistically significant VOC was identified in fecal samples collected from sub-adult wild boar and three were identified in fecal samples from juvenile wild boar. In addition, discriminant function analysis (DFA) was used to build classification models for MTBC prediction in juvenile animals. Using DFA, we were able to distinguish between MTBC-positive juvenile wild boar and MTBC-negative juvenile wild boar using breath VOC or fecal VOC. Based on our results, further research is warranted and should be performed using larger sample sizes, as well as wild boar from various geographic locations, to verify these compounds as biomarkers for MTBC infection in this species. This new approach to detect MTBC infection in free-ranging wild boar potentially comprises a reliable and efficient screening tool for surveillance in animal populations.

## 1. Introduction

The presence of *Mycobacterium tuberculosis* complex (MTBC) in wild swine populations, such as in wild boar (*Sus scrofa*) in Europe [1,2], gives cause for serious concern, not only regarding the health of affected wild swine populations, but also the health of domestic animal species, humans, and other wildlife that share habitats and resources with these populations [3]. Determining presence, prevalence, and dynamics of MTBC in wild swine is extremely important to facilitate early intervention opportunities, as well as to inform ongoing management activities regarding disease reduction and prevention of further disease spread. Development of accurate, efficient, and noninvasive methods to detect MTBC in wild swine would greatly benefit surveillance and disease management efforts. Volatile organic compound (VOC) analysis technology is being explored extensively for its potential to detect tuberculosis in humans as well as in other species, such as cattle (*Bos taurus*) and white-tailed deer (*Odocoileus virginianus*) [4,5,6,7,8,9,10]. Use of these compounds, obtained from various biological samples such as breath and feces, has great potential to fulfill needs for an ideal remote or point-of-care surveillance tool in all species. This study involved collecting breath and fecal VOC from wild boar in Doñana National Park, Andalusia, Spain, to determine whether VOC could aid in distinguishing MTBC-infected boar from noninfected boar.

## 2. Results

### 2.1. Assessment of Volatile Organic Compound Collection

Breath and fecal VOC samples were collected and analyzed using various flow rates and times on three different days in order to assess collection methods that were most effective in capturing VOC from swine (summarized in Table 1 and Appendix A). 

For the samples collected on the first day (Appendix A), good chromatographic signals were obtained for those collected at 1 L/min for 1 min and 1L/min for 2 min, with better shape and slightly better intensity in the case of the 1 L/min for 1 min samples. Practically no chromatographic signals were obtained in the other cases. For the samples collected on the second day (Appendix A), good chromatographic signals were obtained for all samples collected at 1 L/min and 200 mL/min, with better shape and slightly better intensity in the case of the 200 L/min for 1 min samples. For the samples collected on the third day (Appendix A), no chromatographic peaks could be detected. The air samples collected both before and after the breath sample collection did not produce any noticeable signals. 

Next, quality control analysis was performed by measuring the second portion of Tenax material of the samples that produced better chromatographic signals on the first two days of collection (i.e., samples acquired at 1 L/min for 1 min on the first collection day and at 200 mL/min for 1 min on the second collection day). These analyses (shown in Appendix A) revealed that in the case of the sample acquired at 1 L/min for 1 min, no compound was identified in the second portion of the Tenax material; therefore, VOC pre-concentration was adequate in this case. This was in contrast with the sample acquired at 200 mL/min for 1 min, where various compounds were not retained by the first portion of Tenax and passed to the second one. Therefore, these analyses suggested that the best flow rate for the transfer of the breath volatiles to the storage cartridge was 1 L/min for 1 min.

For the fecal samples, the best flow rate for VOC transfer to Tenax was found to be 200 mL/min for 10 min, as evident in Appendix A. The areas of the most abundant chromatographic peaks were notably (visibly or apparently) higher in this case than in the other two cases on the two first collection days and quite similar on the third collection day. 

### 2.2. Animal Disease and VOC Sample Status

Due to dissimilarities in approximately one-third of the breath VOC samples collected from juveniles (8 out of 22 animals) in our field study, a third breath VOC sample was analyzed when the first two samples were found to be dissimilar. In all but one animal, the third sample was similar to one of the first two analyzed; therefore, these two similar samples were used for VOC identification. One animal for whom dissimilarities were found in all three breath samples analyzed was not included in the statistical analysis. Despite the observations above, analysis of 20 breath samples collected from a single animal with sedation and without sedation (data not shown) revealed no indication that breath composition was altered between the collections.

We were able to characterize the MTBC status of 41 of the 57 sampled wild boar. Twenty-two animals were culture-positive for *Mycobacterium bovis* in one or more tissues and thus identified as MTBC-positive. Nineteen animals were identified as MTBC-negative, either with no lesions and culture-negative for *M. bovis*, or with lesions positive for another species of *Mycobacterium* that was not a member of the MTBC. Sixteen animals could not be characterized based on their necropsy and culture results and were therefore were not included for further analysis. Eight additional animals were not included in the statistical analysis for the sake of consistency between groups as they did not have data pertaining to both breath and fecal VOCs. Ultimately, 15 MTBC-positive and 18 MTBC-negative wild boar were used for the final analysis (Table 2). 

### 2.3. Significant Volatile Organic Compounds

In the breath of adult wild boar, one compound (O-cymene) was found to be present at significantly higher levels in MTBC-negative animals than in MTBC-positive animals. No significant compounds were identified in the breath of sub-adults. In juveniles, 14 compounds were found at significantly higher levels in the breath of MTBC-positive animals compared with MTBC-negative animals. These statistically significant compounds are presented in Table 3. The comparative abundance of these compounds in the breath samples of all animals from each group is shown in Figure 1. 

Regarding feces, no significant compounds were identified in adults. In sub-adults, one compound was found to be significantly more abundant in MTBC-negative animals. Three compounds were significantly more abundant in the feces of MTBC-negative juveniles. The significant fecal VOC for MTBC in wild boar identified in this study are presented in Table 4. The comparative abundance of these compounds in the feces of all animals from each group is shown in Figure 2. 

### 2.4. Classification Models for MTBC Prediction

Discriminant function analysis (DFA) classification models for MTBC prediction were built for the juvenile animals, where more than one significant compound was identified. To prevent model overtraining, a small number of VOC were selected to build the classification models. The DFA models were built with three breath VOC (JBVOC 11, JBVOC 13, and JBVOC 14, shown in Table 3) and two fecal VOC (JFVOC 01 and JFVOC 03, shown in Table 4), as well as the receiver operating characteristic (ROC) curves built with the first canonical variable (CV1) of these classification models, which are shown in Figure 3. 

The selection of these compounds was based on their low p-values in the statistical analysis. The prediction accuracy, sensitivity, and specificity were 93.3%, 100%, and 90%, respectively, for the classification model built with the breath VOC, and 86.7%, 100%, and 80%, respectively, for the classification model built with the fecal VOC. The area under the ROC curve (AUC) analysis yielded 94% and 100% accuracy for the classification models built with the breath and fecal VOC, respectively. 

## 3. Discussion

This is the first report of identification of volatile organic compounds obtained from the breath and feces of wild boar in order to distinguish between MTBC-positive and MTBC-negative boar. This novel approach to detect MTBC infection in free-ranging wild boar has the potential to be a reliable and efficient complement as a diagnostic screening tool for disease surveillance in this species, as well as in other affected wild animal populations [5,11,12]. We identified significant VOC in the breath and feces of MTBC-positive and MTBC-negative wild boar of three age classes from Doñana National Park, Spain. When applying classification models to the juvenile animals, for whom several significant VOC were identified, we were able to distinguish between MTBC-positive juvenile wild boar and MTBC-negative juvenile wild boar using breath VOC or fecal VOC. 

During mycobacterial infection, the predominant cellular immune responses lead to cellular destruction via release of cytokines and highly reactive species resulting in tissue damage related to oxidative stress. Overproduction of highly reactive oxygen intermediates and reactive nitrogen radicals leads to disruptive cellular damage to tissue membranes, cleaving of proteins, accumulation of lipids, mainly polyunsaturated fatty acids (PUFA), and excessive mitochondrial respiration. In animals, reactive oxygen molecules can be produced from arachidonic acids (such as PUFA) by lipoxygenase enzymes during disease pathogenesis [13]. The majority of compounds identified as significant in juvenile wild boar breath were alkane derivatives (nine compounds), followed by aromatic (benzene) derivatives (three compounds), one alkene, and one ester. Most of the breath VOC identified in juvenile wild boar exhibited enhanced concentrations in MTBC-positive animals, with the exception of three compounds (JBVOC 03, JBVOC 05, and JBVOC 09), whose concentrations were diminished in the breath of MTBC-positive animals. Alkanes and alkane derivatives were previously observed in human breath as VOC products of oxidative stress correlated to active pulmonary tuberculosis, while benzene derivatives were identified as volatile metabolites of *M. tuberculosis* in vitro [6,7]. Among the alkane derivatives, JBVOC 10 to JBVOC 14 were present in the breath of all MTBC-positive animals, but in the breath of only 30% of MTBC-negative animals.

Alkane derivatives such as 3-methylpentane, decane, and heptacosane are thought to originate as byproducts of lipid peroxidation during MTBC pathogenesis [13]. A structural isomer of 3-methylpentane, 2-methylpentane, was also found in the breath samples of mice infected with *M. bovis*, but significant difference was not shown compared to uninfected controls [14]. On the other hand, although the concentration of decane in MTBC-infected animals depicts downregulation, it might be readily adsorbed onto cell membranes upon production and follow further metabolic bioconversion to lower the molecular weight VOC and/or alkane derivatives. Branched alkanes, such as 5-butyl-5-ethylheptadecane, 11-decyl-tetracosane, 11-(1-ethylpropyl)-heneicosane, and 3-ethyl-5-(2-ethylbutyl)-octadecane, might originate from the excessive peroxidation of long-chain polyunsaturated fatty acids during oxidative damage of cell membranes. 

The aromatic derivatives α-methylstyrene and 1,3-bis(1,1-dimethylethyl)-benzene were significantly increased in the breath samples of MTBC-positive wild boar, whereas 2,5-bis(1,1-dimethylethyl)-phenol exhibited increased concentration in the breath samples of MTBC-negative animals. It is possible that gut bacteria produce phenolic compounds from the metabolism of aromatic amino acids. Although aromatic compounds were linked to diseases of the respiratory tract, their appearance in exhaled breath often corresponds to exogenous origins and are widespread in the body, therefore their role as specific biomarkers for infectious diseases seems to be doubtful [15,16]. 

The alkene derivative 4,6,8-trimethyl-1-nonene was upregulated in breath samples of MTBC-positive wild boar, which might be attributed to increased peroxidation of unsaturated fatty acids and chain cleavage during oxidative stress. The branched alkane peroxide 2,5-dimethylhexane-2,5-dihydroperoxide was increased in breath samples of MTBC-positive animals, which could be associated with infection-induced inflammation and the occurrence of lesions in the intestinal epithelia. The levels of MTBC-induced disease in the intestinal tracts of the wild boar in this study were relatively low based on lesions observed in the mesenteric lymph nodes; however, parasitic infections and other sources of intestinal compromise commonly occur in this population and may have played a role in the production of this compound. 

Regarding other compounds noted as significant between positive and negative juvenile wild boar, ester molecules such as acetic acid and methyl ester might be produced during the acid-catalyzed reaction of acetic acid with primary alkanols and could be a potential indicator of altered activity and metabolic changes after the binding of a pathogen to an epithelial cell. Carboxylic ester molecules were also found in breath samples of *M. bovis* BCG-infected mice [14]. The alkane derivative trichloromethane was found at higher concentrations in breath samples of MTBC-negative animals, which could be attributed to its exogenous origin in addition to the endogenous origin reported in human breath [17,18]. Breath samples collected from adult wild boar revealed one branched aromatic hydrocarbon, O-cymene, as significant. This compound may be related to endogenous oxidation during aromatic metabolism, where aromatic hydrocarbons are readily adsorbed onto cell membranes after exogenous ingestion though skin and/or from food [19]. It is important to note that alkane and aromatic derivatives were reported to be prevalent in outdoor animal farms, such as pigs and cattle [20].

As an alternative to breath exhalation, other compounds are biotransformed into polar soluble metabolites and can be excreted through feces [21]. All fecal VOC identified in the present study in wild boar showed increased levels in the feces of MTBC-negative animals, most of which were aromatic compounds. The branched aromatic hydrocarbon 10,18-bisnorabieta-8,11,13-triene in sub-adult wild boar might correspond to an exogenous origin related to food ingestion. Toluene and 2,6-bis(1,1-dimethylethyl)-4-(1-methylpropyl)-phenol, which were identified in juvenile wild boar, are thought to be derived from the metabolic oxidation of aromatic proteins by members of cytochrome P450 enzymes during mycobacterial infection. Lower concentrations of these aromatic VOC excreted in the feces of MTBC-positive animals might be related to their polar functionality, thereby making them soluble in body fluids and allowing excretion through other body fluids. Of note, 2,6-bis(1,1-dimethylethyl)-4-(1-methylpropyl)-phenol) was not found in the feces of any MTBC-positive animal, but was also absent in the feces of two MTBC-negative animals out of the 10 included in the statistical analysis. Acetone, the only ketone identified as significant in juvenile wild boar, might be attributed to increased metabolic activity during mycobacterial infection.

The analysis of two breath samples collected from each animal revealed consistent similarities with each other for all adult and sub-adult animals, but only for approximately two-thirds of the juvenile animals. The dissimilarities between the two breath samples of these juvenile animals could be attributed to the fact that they were not anesthetized during breath sampling. They were manually restrained without sedation, which may have produced stress responses during breathing that altered the breath composition. Nevertheless, the measurement of a third breath sample of these animals revealed consistent similarities with one of the first two samples analyzed for all but one animal. No differences were observed in the composition of the two fecal VOC samples for all animals, indicating that the stress induced in the juvenile animals during manual restraint did not produce an immediate effect on fecal VOC composition. 

Our study demonstrated the occurrence of a certain set of VOC associated with the presence of MTBC infection in wild boar. However, our small sample sizes likely introduced limitations to bear in mind when considering them as potential biomarkers of MTBC pathogenesis in wild boar. Further studies involving larger numbers of MTBC-positive and MTBC-negative wild boar of different age classes and various geographic locations are needed to determine whether these compounds, and perhaps others, could be seen as definitive in differentiating MTBC-infected from noninfected animals.

## 4. Methods

### 4.1. Assessment of Volatile Organic Compound Collection

The most optimal methods for VOC collection from breath and feces in swine were determined prior to collection in the field. We collected samples from an MTBC-negative, captive-raised, feral pig at the United States Department of Agriculture (USDA), Animal and Plant Health Inspection Service (APHIS)/Colorado State University (CSU), Wildlife Research Facility in Fort Collins, CO, USA, that was participating in a related study approved by the CSU Institutional Animal Care and Use Committee under protocol #14-5367A. The animal was manually restrained in a Panepinto Sling® (Panepinto and Associates, Masonville, CO, USA). VOC were collected from breath as described in [22]. Briefly, a modified equine nebulization mask (Aeromask, Trudell Medical International, London, ON, Canada) was placed over the animal’s muzzle. Breath was drawn via a vacuum pump (AirChek XR5000, SKC Inc., Eighty Four, PA, USA) through a 5 cm section of Tygon tubing (3/8 inches outer diameter, 1/4 inches inner diameter) (Saint-Gobain Performance Plastics, Akron, OH, USA) emerging from the mask, followed by a three-piece bioaerosol cassette (SKC Inc., Eighty Four, PA, USA) containing a 37 mm, 0.22 μm polytetrafluoroethylene (PTFE) filter (Tisch Scientific, North Bend, OH, USA) and a 37 mm cellulose pad (SKC Inc. Eighty Four, PA, USA). This section was followed by a 20 cm section of Tygon tubing connecting a glass cartridge containing sorbent material (Tenax; Tenax TM; Sigma Aldrich, St. Louis, MO, USA), and followed by a 20 cm section of Tygon tubing leading to the pump (Figure 4). Breath samples were collected at a rate of 100 mL/min for 1 min, 100 mL/min for 2 min, 200 mL/min for 1 min, 200 mL/min for 2 min, 1 L/min for 1 min, and 1 L/min for 2 min. One sample under each rate condition was collected per day on 3 different days during one week. Therefore, three samples were obtained for each condition. 

Fecal samples were manually collected from the rectum of the pig, of which 5 g was placed in a 125 mL wide-mouth glass specimen jar with septum in the lid (I-CHEM 200; VWR International, Radnor, PA, USA). Two 12-gauge stainless steel needles were inserted into the jar with one of the needle points positioned just above the fecal sample but not contacting the sample. Headspace air was drawn from the jar as it was from the mask by attaching the tubing to the hub of the needle above the sample. The second needle allowed air flow to occur above the sample (Figure 4). Fecal headspace air was collected at 200 L/min for 2 min, 200 L/min for 5 min, and 200 L/min for 10 min. Three samples were obtained under each condition on the same three different days as for the breath collection. 

Additionally, air samples were collected on every collection day, before and after sampling of breath and fecal VOC, to serve as controls. Air samples for breath VOC controls were collected at 1 L/min for 1 min in the same location and with the same collection apparatus with a clean mask as the breath VOC. Air samples associated with fecal VOC collection were collected at 1 L/min for 1 min using the entire fecal collection apparatus with a clean jar in the same location as the fecal VOC collections. All Tenax TM cartridges were stored at 4 ℃ until shipment to and analysis at Rovira i Virgili University (URV), Tarragona, Spain. 

### 4.2. Volatile Organic Compound Analysis of Assessment Samples

Both regions of the Tenax TM cartridges were analyzed using gas chromatography–mass spectrometry (GC–MS). The sorbent cartridges contained two regions of Tenax material, one of which was aimed at preconcentrating the volatiles, while the second region was used for quality control (i.e., to verify if the volatiles were adequately adsorbed by the Tenax material from the first region and did not pass to the second region). The two regions of Tenax were separately analyzed using GC–MS combined with a headspace sampler system (LECO Corporation, Saint Joseph, MI, USA). 

The Tenax material was introduced in 20 mL sealed vials compatible with the headspace system. Samples were heated at 250 ℃ for 10 min to release volatiles from the Tenax material. The released VOC were automatically transferred to the GC–MS for chromatographic analysis. The injection port was configured in the splitless injection mode. The GC oven temperature profile used was (i) 40 ℃, hold for 5 min, (ii) ramp of 5 ℃/min until 140 ℃, (iii) ramp of 15 ℃/min until 300 ℃, and (iv) hold for 5 min at 300 ℃. Volatile organic compounds were chromatographically separated using an HP-5MS capillary column, 30 m × 0.32 mm, 0.25 µm film thickness, with 5% phenyl methyl siloxane, (Agilent Technologies, Spain). 

### 4.3. Wild Boar Field Study

Volatile organic compounds were collected from 57 wild boar captured at Doñana National Park (37°0′ N, 6°30′ W) in the provinces of Huelva and Seville, Andalusia, Spain. Study procedures were approved by the Animal Experiment Committee of Castilla-La Mancha University (PR-2015-03-08), Spain, and were designed and developed by scientists (B and C animal experimentation categories) approved by the Spanish Ethics Committee in accordance with European Commission Directive 86/609/EEC for animal handling and experiments. Wild boar at Doñana National Park, ranging in age from juvenile to adult of both sexes, were captured using cage and corral traps as described by Barasona et al., 2014 [23]. Once an animal was found in a trap, depending on age and size of the animal, the boar was anesthetized with a combination of tiletamine–zolazepam (Zoletil® 100 mg/mL, Virbac, France, target dose 3 mg/kg) and medetomidine (Medetor®, Virbac, France, target dose 0.05 mg/kg), both administered intramuscularly in the lateral thigh region with 5 mL darts (Telinject®, Römerberg, Germany) using a 14 mm diameter blowpipe (Telinject®, Römerberg, Germany), after visually estimating the weight of each animal in the trap [24]. After sedation, wild boar were removed from the traps and blindfolded for breath and fecal collection. Smaller animals (i.e., juveniles) were manually restrained without sedation.

Volatile organic compounds were collected as described above (Figure 4), with the one exception that in some of the wild boar, the fecal sample was collected postmortem when no feces could be obtained per rectum during handling. Sampling parameters for fecal and breath collection were chosen based on the quality of signals observed in the chromatograms obtained in the optimization process. Based on optimization results (see the Results section), breath samples were collected at a rate of 1 L/min for 1 min, whereas fecal sample headspace volatiles were collected at a rate of 200 mL/min for 10 min. Fecal samples were stored at 4 ℃ until VOC collection, usually within 8 hours but sometimes within 24 hours if same-day VOC collection was not possible. For all but one animal, four breath VOC samples were collected from each animal, while for all animals, four Tenax cartridges were collected from each fecal sample. For purposes of comparing breath VOC collected from sedated and nonsedated animals, 10 breath VOC samples were collected from one individual (MTBC-positive juvenile female) before sedation and after sedation. All Tenax TM cartridges were stored at 4 ℃ until transport to and analysis at URV. 

Once breath VOC and fecal samples were collected, the animal was euthanized via captive bolt and exsanguination as part of the Doñana National Park health-monitoring program. Wild boar were necropsied and blood, lung lobes (paired when applicable; diaphragmatic, cranial, cardiac, accessory), and lymph nodes (paired; mandibular, mediastinal, bronchial, mesenteric) were taken for culture. Any other tissue displaying lesions compatible with MTBC infection were also collected. 

Tissue preparation and culture were carried out as described by Gortázar et al. (2008) [25]. Briefly, 1–4 g of tissue was decontaminated with N-acetyl-l-cysteine and NaOH, concentrated by centrifugation and inoculated in solid Löwenstein-Jenssen medium with pyruvate. Identification to species level was performed with the GenoType^®^ MTBC system (Hain Lifescience GmbH, Nehren, Germany). All microbiology was done at the Virgen del Rocío Hospital in Seville, Spain, in a laboratory meeting biosecurity level 3 requirements. 

### 4.4. Volatile Organic Compound Analysis of Field Samples

VOC samples were analyzed with a quadrupole time-of-flight gas chromatography/mass spectrometry (GC/Q-TOF) system (Agilent Technologies, Santa Clara, CA, USA), which is state-of-the-art in untargeted GC–MS analysis. Only the first region of the Tenax material, aimed at volatile preconcentration, was analyzed by GC/Q-TOF. Tenax was introduced in 20 mL sealed glass vials provided with a septum, which were heated at 100 ℃ for 20 min inside an oil bath placed on a temperature-controlled hotplate for thermal desorption of the VOC trapped by the Tenax during sample collection. Solid-phase micro-extraction (SPME) with divinylbenzene/carboxene/polydimethylsiloxane (DVB/CAR/PDMS) fiber, introduced in the headspace formed by the released volatiles inside the sealed vials during VOC thermal desorption process, was used to capture and concentrate the released VOC and to inject them into the GC/Q-TOF splitless port for analysis. The GC oven temperature profile was (i) 35 ℃, hold for 5 min, (ii) ramp of 4 ℃/min until 150 ℃, (iii) ramp of 10 ℃/min until 250 ℃, and (iv) hold for 2 min at 250 ℃. The chromatographic column employed was an HP-5MS capillary column, 30 m × 0.32 mm, 0.25 µm film thickness, with 5% phenyl methyl siloxane (Agilent Technologies, Spain). 

Acquired chromatograms were analyzed using Unknown Analysis software (Qualitative and Quantitative Analysis B.07.00, Unknown Analysis, Agilent Technologies, Santa Clara, CA, USA) operated in manual mode. Compound identification was achieved using the NIST 14 mass spectral library.

### 4.5. Data Analysis

Analyses were divided into three age classes, namely, adults (>2 years), sub-adults (12–24 months), and juveniles (<12 months). Mean VOC values were obtained from measurements of two Tenax cartridges. If dissimilarities were noted in the VOC samples from the first two analyzed Tenax cartridges, a third VOC sample was analyzed and two similar cartridges of the three were then chosen for statistical analysis. If there were additional dissimilarities noted in the third sample, that animal was removed from the analysis.

For each animal age and each kind of VOC sample analyzed in this study, statistical analysis was performed on compounds that were found in more than 80% of the animals. Tentative biomarkers were identified by applying a two-tailed statistical t-test for two samples assuming unequal variance (using QI Macros 2017 toolbox for Excel) with an α value of 0.05 to determine statistically significant differences. A second statistical analysis based on animal sex and location, performed on the initial list of significant VOC, disregarded the compounds that could be affected by these factors.

Discriminant function analysis was used to build classification models for MTBC prediction [26]. DFA is a linear supervised pattern-recognition algorithm that finds new orthogonal variables called canonical variables as a combination of the input variables (compound abundance in this study) to minimize the distance between samples from the same group and maximize the distance between samples from different groups (MTBC-positive and MTBC-negative animals in this study). The prediction parameters of the DFA models were estimated through leave-one-out cross-validation. The accuracy, sensitivity, and specificity of prediction were calculated in function of true positive (TP), true negative (TN), false positive (FP), and false negative (FN) values, as follows: accuracy = (TP + TN)/(TP + TN + FP + FN); sensitivity = TP/(TP + FN); specificity = TN/(TN + FP). The prediction accuracy of the DFA classification models built was assessed by calculating the AUC of the ROC curve built with the first canonical variable, which is the most discriminative in the DFA models aiming to classify the two groups [27]. 

## Figures and Tables

**Figure 1 pathogens-09-00346-f001:**
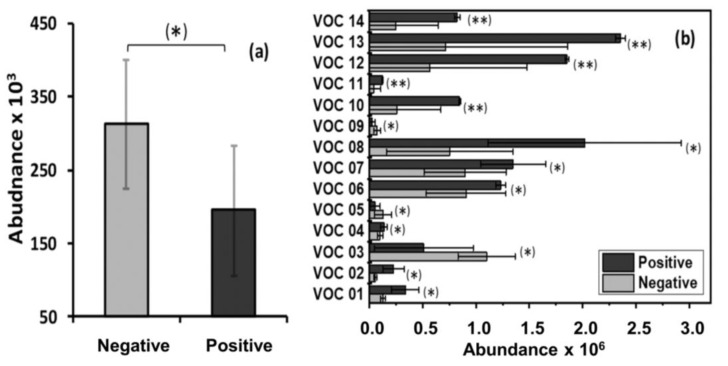
Mean values of the abundance (area under the chromatographic peak) of the significant breath VOC found in this study for a) adult wild boar and b) juvenile wild boar. Error bars represent the standard error of the mean. * Statistically significant difference (*p* < 0.05) between MTBC-positive and MTBC-negative groups; ** statistically significant difference (*p* < 0.01) between MTBC-positive and MTBC-negative groups.

**Figure 2 pathogens-09-00346-f002:**
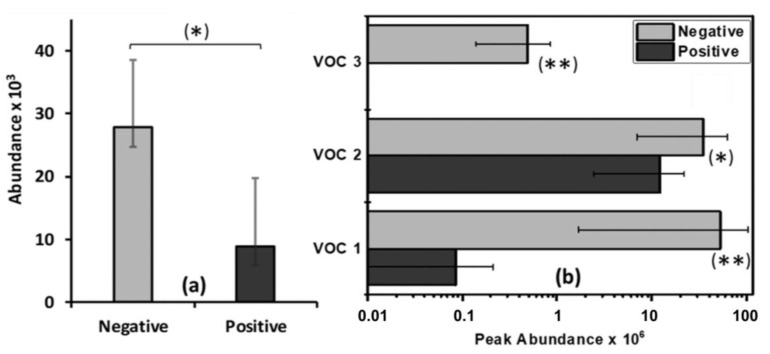
Mean values of the abundance (area under the chromatographic peak) of the fecal VOC biomarkers found in this study for (**a**) sub-adult wild boar and (**b**) juvenile wild boar. The error bars represent the standard error of the mean. * Statistically significant difference (*p* < 0.05) between MTBC-positive and MTBC-negative groups; ** statistically significant difference (*p* < 0.01) between MTBC-positive and MTBC-negative groups.

**Figure 3 pathogens-09-00346-f003:**
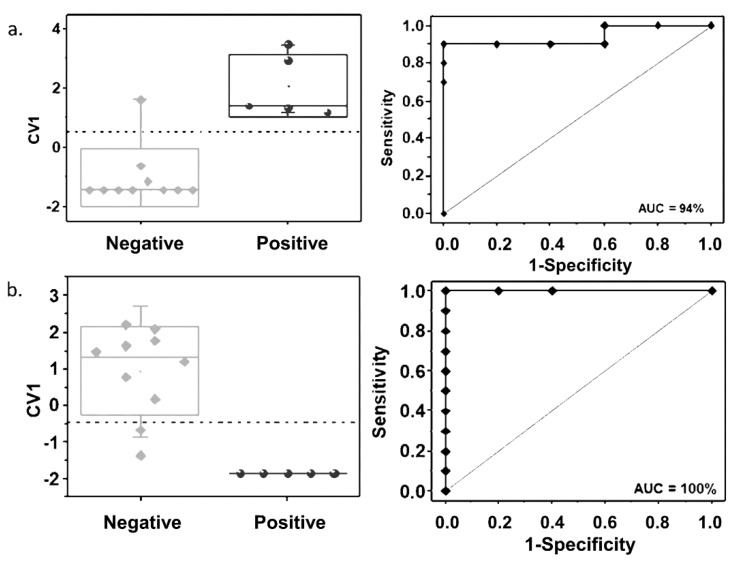
Classification between MTBC-positive and MTBC-negative animals achieved by the discriminant functional analysis (DFA) models built with (**a**) three breath VOC biomarkers (JBVOC 11, JBVOC 13, and JBVOC 14, shown in Table 3) and (**b**) two fecal VOC biomarkers (JFVOC 01 and JFVOC 03, shown in Table 4). Left panels: Box plots of the first canonical variable (CV1) of the DFA models. Each animal is represented by one point in the box plots. The standard deviation of the CV1 values is represented by the error bars, while the boxes represent the 95% confidence interval and the dashed lines represent the threshold classification line between the MTBC-positive and MTBC-negative groups. Right panels: receiver operating characteristic (ROC) curves built with the CV1 of the DFA models.

**Figure 4 pathogens-09-00346-f004:**
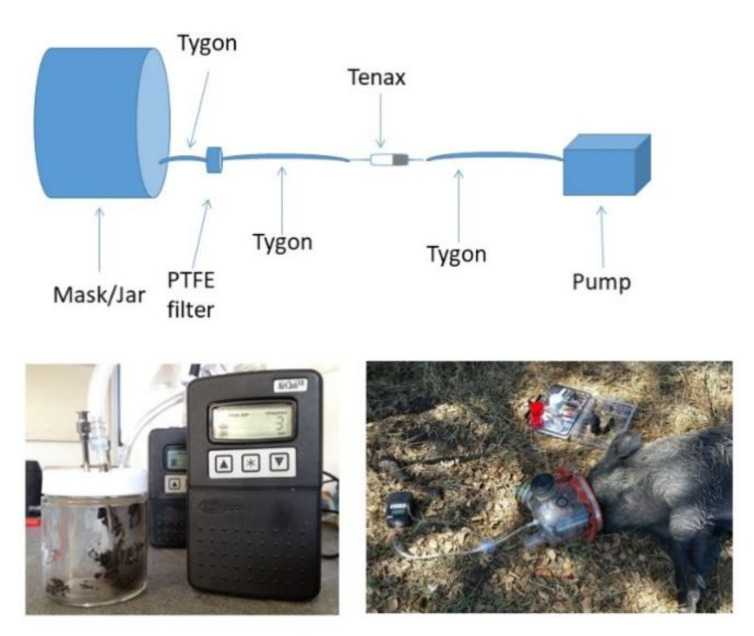
Illustration of VOC collection apparatus and images of fecal sample collection and breath sample collection from wild boar in Doñana National Park, Spain.

**Table 1 pathogens-09-00346-t001:** Breath and fecal volatile organic compound (VOC) samples that produced good chromatographic signals on each collection day. Breath VOC samples were collected from one animal over three days at a rate of 100 mL/min for 1 min, 100 mL/min for 2 min, 200 mL/min for 1 min, 200 mL/min for 2 min, 1 L/min for 1 min, and 1 L/min for 2 min. Fecal headspace air was collected from one animal over three days at a rate of 200 mL/min for 2 min, 200 mL/min for 5 min, and 200 mL/min for 10 min. Three samples were obtained under each condition.

		Day		
Sample Source	Flow Rate	1	2	3
Breath	100 mL/min for 1 min			
	100 mL/min for 2 min			
	200 mL/min for 1 min		√	
	200 mL/min for 2 min		√	
	1L/min for 1 min	√	√	
	1L/min for 2 min	√	√	
Feces	200 mL/min for 2 min			
	200 mL/min for 5 min			
	200 mL/min for 10 min	√	√	√

**Table 2 pathogens-09-00346-t002:** Number of *Mycobacterium tuberculosis* complex (MTBC)-positive and MTBC-negative wild boar included in the final volatile organic compound analysis, specifying gender and age class.

Age	Nº of Animals			
	MTBC-Negative		MTBC-Positive	
	Male	Female	Male	Female
Adult	2	4	4	2
Sub-adult	-	2	3	1
Juvenile	4	6	3	2
Total	6	12	10	5

**Table 3 pathogens-09-00346-t003:** Significant breath volatile organic compounds identified in MTBC-positive and MTBC-negative adult and juvenile wild boar.

Age	VOC Number	VOC	Chemical Family	*p*-Value
Adult	ABVOC ^1^ 01	O-cymene	Aromatic	0.045
Juvenile	JBVOC ^2^ 01	Acetic acid, methyl ester	Ester	0.022
	JBVOC 02	3-methylpentane	Alkane	0.018
	JBVOC 03	Trichloromethane	Alkane Derivative	0.047
	JBVOC 04	α-methylstyrene	Aromatic	0.046
	JBVOC 05	Decane	Alkane	0.045
	JBVOC 06	4,6,8-trimethyl-1-nonene	Alkene	0.024
	JBVOC 07	1,3-bis(1,1-dimethylethyl)-benzene	Aromatic	0.033
	JBVOC 08	2,5-dimethylhexane-2,5-dihydroperoxide	Alkane Derivative	0.036
	JBVOC 09	2,5-bis(1,1-dimethylethyl)-phenol	Aromatic Derivative	0.012
	JBVOC 10	Heptacosane	Alkane	0.001
	JBVOC 11	5-butyl-5-ethylheptadecane	Alkane	0.003
	JBVOC 12	11-decyl-tetracosane	Alkane	0.002
	JBVOC 13	11-(1-ethylpropyl)-heneicosane	Alkane	0.001
	JBVOC 14	3-ethyl-5-(2-ethylbutyl)-octadecane	Alkane	0.001

^1^ ABVOC: adult breath VOC; ^2^ JBVOC: juvenile breath VOC.

**Table 4 pathogens-09-00346-t004:** Significant fecal volatile organic compounds identified in MTBC-positive and MTBC-negative sub-adult and juvenile wild boar.

Age	VOC Number	VOC	Chemical Family	*p*-Value
Sub-Adult	SAFVOC ^1^ 01	10,18-bisnorabieta-8,11,13-triene	Aromatic	0.048
Juvenile	JFVOC ^2^ 01	Acetone	Ketone	0.009
	JFVOC 02	Toluene	Aromatic	0.041
	JFVOC 03	2,6-bis(1,1-dimethylethyl)-4-(1-methylpropyl)-phenol	Aromatic	0.002

^1^ SAFVOC: sub-adult fecal VOC; ^2^ JFVOC: juvenile fecal VOC.

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
