# Peer review of "Evaluation of Volatile Organic Compounds Obtained from Breath and Feces to Detect Mycobacterium tuberculosis Complex in Wild Boar (Sus scrofa) in Doñana National Park, Spain"

_pathogens, 2020, doi:10.3390/pathogens9050346_

Round 1

Reviewer 1 Report

MTBC are very broad in terms of antigens, host specificity and pathogenesis. This manuscript represent volatile compounds captured in the breath and feces of wild boar. Does the authors think that technique is superior than sampling sputum or feces for bacterial growth? Or better than PCR, Cytokines profiling?

Are these volatile compounds specific to wild boar? The sensitivity of the technique is based on models and are not 100%. It would be interesting to gather information from higher sample size and also compare with samples from human or other animals.

Author Response

MTBC are very broad in terms of antigens, host specificity and pathogenesis. This manuscript represent volatile compounds captured in the breath and feces of wild boar. Does the authors think that technique is superior than sampling sputum or feces for bacterial growth? Or better than PCR, Cytokines profiling?

Thank you for your question.  One of the main goals of this technique is to eventually utilize it as a hands off or remote screening/detection tool, a quality that would be beneficial in many settings, including in the context of wildlife disease investigations.  It could increase the efficiency of detecting presence of infection in a single animal or a groups of animals, or humans, as current testing techniques can take several days, and sometimes involving multiple handling events, to months, in the case of culture.  It is not seen as something to replace culture or PCR of biological specimens, or cytokine profiling, as these techniques are still vital for confirmatory diagnosis, environmental studies, immunological studies, and treatment response monitoring.  

Are these volatile compounds specific to wild boar? The sensitivity of the technique is based on models and are not 100%. It would be interesting to gather information from higher sample size and also compare with samples from human or other animals.

The reviewer is absolutely correct in this statement. We do not know at this point whether the VOC are specific to wild boar, or the bacterium, or both.  In addition, individuals will also have unique responses to infection that may produce different profiles.  Much higher sample sizes in many different species are needed to elucidate differences in order to advance these techniques..  Animals and humans with related infections by organisms within the genus Mycobacterium must be investigated and compared, as well as animals and humans with co-infections with other bacteria or viruses. In addition, these samples must come from different environments as that can play a role as well.

Reviewer 2 Report

Dear Authors,

      I have reviewed the manuscript entitled "Evaluation of volatile organic compounds obtained from breath and feces to detect Mycobacterium tuberculosis complex in wild boar (Sus scrofa) in Doñana National Park, Spain and I found it well written and of interest for the scientific community. The article presents the development of a potential non-invasive method for screening MTBC in wild swine populations that could be improved, adapted and extended for other wild animals. 

      Even if the research is mainly well presented, I still have some observations regarding the manuscript:

  1. Please write all species names in italics: (e.g. Mycobacterium tuberculosis complex - line 57).
  2. Line 74: Please change "Fecal" to "fecal".
  3. Lines 93, 100, 317: I believe "L" should be corrected to "mL". Was the fecal headspace air collected at 200 mL/min for 2 min, 5 min and 10 min, respectively or at 200 L/min? Please correct using the correct data.
  4. Why in Figure 2c the color code was changed? Why not use the same color code in all 3 images presented in figure 2 (yellow for 200mL/min for 2 min, green for 200 mL/min for 5 min and blue for 200 mL/min for 10 min)? It's confusing for the reader.
  5. Table 3: I would replace "alkane" with "alkane derivative" for the following compounds: trichloromethane (it belongs to the halogenated derivatives chemical class), 2,5-dimethylhexane-2,5-dihydroperoxide (it belongs to the hidroperoxides class) and "aromatic" to "aromatic derivative" for 2,5-bis(1,1-dimethylethyl)-phenol (it belongs to the phenols chemical class).
  6. Line 247: I would change "alkane" to "alkane derivative".
  7. Lines 266-267: Please correct "2,6-bis(1,1-dimethylethyl)-4-(1-methylpropyl)-)" (partial name of the compound) to "2,6-bis(1,1-dimethylethyl)-4-(1-methylpropyl)-phenol".
  8. Lines 355 and 358: Please change "boar" (singular) to "boars" (plural).

Author Response

Even if the research is mainly well presented, I still have some observations regarding the manuscript:

Thank you for your comments.  We have responded to them below.

  1. Please write all species names in italics: (e.g. Mycobacterium tuberculosis complex - line 57). Corrected as per reviewer’s instructions

  1. Line 74: Please change "Fecal" to "fecal". Corrected as per reviewer’s instructions
  2. Lines 93, 100, 317: I believe "L" should be corrected to "mL". Was the fecal headspace air collected at 200 mL/min for 2 min, 5 min and 10 min, respectively or at 200 L/min? Please correct using the correct data. Corrected as per reviewer’s instructions

  1. Why in Figure 2c the color code was changed? Why not use the same color code in all 3 images presented in figure 2 (yellow for 200mL/min for 2 min, green for 200 mL/min for 5 min and blue for 200 mL/min for 10 min)? It's confusing for the reader. We will try to make changes to this figure in order to decrease the confusion caused by the color coding

  1. Table 3: I would replace "alkane" with "alkane derivative" for the following compounds: trichloromethane (it belongs to the halogenated derivatives chemical class), 2,5-dimethylhexane-2,5-dihydroperoxide (it belongs to the hidroperoxides class) and "aromatic" to "aromatic derivative" for 2,5-bis(1,1-dimethylethyl)-phenol (it belongs to the phenols chemical class). Corrected as per reviewer’s instructions

  1. Line 247: I would change "alkane" to "alkane derivative". Corrected as per reviewer’s instructions

  1. Lines 266-267: Please correct "2,6-bis(1,1-dimethylethyl)-4-(1-methylpropyl)-)" (partial name of the compound) to "2,6-bis(1,1-dimethylethyl)-4-(1-methylpropyl)-phenol". Corrected as per reviewer’s instructions

  1. Lines 355 and 358: Please change "boar" (singular) to "boars" (plural). Boar can be both singular and plural and the authors prefer to use boar for both.

 [NP-A1]

Reviewer 3 Report

This study is interesting and will be the first publication describing how Volatile Organic Compounds are obtained and identified from breath and feces in boar and more importantly how this novel approach can help to detect animals infected by mycobacterium tuberculosis complex.

I have 2 minor comments:

1) Graphic quality in Figure 1 is suboptimal (blurry and difficult to read) - Please improve the graph quality

2) Why were the material and methods described after the discussion? It makes the study hard to follow and should be changed.

Author Response

Thank you for your comments.  We have attempted to clarify some of the wording in the results section in response to the comment that the results needed improvement.  However there were no specific comments related to the results except for Figure 1. We will attempt to improve Figure 1 so that it is more clear and readable.

1) Graphic quality in Figure 1 is suboptimal (blurry and difficult to read) - Please improve the graph quality 

The authors will work on improving this image.

2) Why were the material and methods described after the discussion? It makes the study hard to follow and should be changed.

This formatting is per Pathogens journal template